# A Pilot Study on the Influence of Diaphragmatic Function on Iliopsoas Muscle Activity in Individuals with Chronic Ankle Instability

**DOI:** 10.3390/muscles4020016

**Published:** 2025-05-19

**Authors:** Takumi Jiroumaru, Shun Nomura, Yutaro Hyodo, Michio Wachi, Junko Ochi, Nobuko Shichiri, Takamitsu Fujikawa

**Affiliations:** 1Department of Physical Therapy, School of Health Sciences, Bukkyo University, Kyoto 604-8418, Japan; y-hyodo@bukkyo-u.ac.jp (Y.H.); m-wachi@bukkyo-u.ac.jp (M.W.); ochi@bukkyo-u.ac.jp (J.O.); fujikawa@bukkyo-u.ac.jp (T.F.); 2Department of Rehabilitation, Kanazawa Orthopaedic and Sports Medicine Clinic, Ritto 520-3016, Japan; sn610101@gmail.com; 3Department of Occupational Therapy, School of Health Sciences, Bukkyo University, Kyoto 604-8418, Japan; shichiri@bukkyo-u.ac.jp

**Keywords:** iliopsoas, chronic ankle instability, diaphragm, surface electromyography, breath-holding conditions, hip flexion torque

## Abstract

This study examined the impact of different breath-holding conditions on iliopsoas and other hip flexor muscle activity in individuals with chronic ankle instability (CAI). It has been hypothesised that impaired diaphragmatic function influences iliopsoas activation, potentially contributing to motor control deficits in patients with CAI. Eleven adults with a history of chronic ankle sprain participated in this study. Maximal isometric hip flexion was assessed under three breath-holding conditions: end-expiration, end-inspiration, and the intermediate state. Surface electromyography was used to record the muscle activity of the iliopsoas, rectus femoris, sartorius, and tensor fasciae latae, while the peak hip flexion torque was measured using an isokinetic dynamometer. Under the end-inspiration condition, iliopsoas activity on the affected side was significantly lower than that on the control side (*p* < 0.05). However, no significant differences were observed between the affected and control sides in the activity of the other hip flexor muscles or the peak hip flexion torque across breath-holding conditions. This study highlights the association between reduced iliopsoas activity during end-inspiration and compromised diaphragmatic function in patients with CAI. Future research should explore dynamic movement tasks and larger sample sizes to elucidate neuromuscular mechanisms further and refine rehabilitation strategies.

## 1. Introduction

Ankle sprains are among the most common musculoskeletal injuries, and several cases involve damage to the lateral ligament complex [1,2]. While appropriate treatment and rehabilitation during the acute phase typically restore function, inadequate management or repeated injuries can lead to chronic ankle instability (CAI) [3,4]. CAI is associated with sensory deficits and altered muscle activation patterns, ultimately impairing neuromuscular control [5,6,7]. These impairments may increase the risk of falls and limit sports performance [5,6,7]. Although rehabilitation programmes targeting ankle strength and proprioception have been implemented, many individuals with CAI continue to experience functional deficits and recurrent episodes of instability [8,9,10,11]. These persistent impairments may be attributed to residual neuromuscular deficits, altered proprioception, and disruptions in central neuromuscular coordination, including compromised core stability. Recent studies suggest that although balance and strength training can improve functional outcomes [8], they may not adequately address the role of proximal muscle activation, particularly in the diaphragm and hip flexors. This highlights the need for rehabilitation strategies that incorporate local joint mechanics and central neuromuscular control to enhance functional recovery in individuals with CAI.

Recent epidemiological investigations have underscored the substantial impact of ankle sprains on public health, noting that ankle sprains account for a significant percentage of emergency department visits worldwide [12]. The high prevalence of ankle sprains across various age groups and activity levels highlights the urgent need for a deeper understanding of both the acute injury mechanisms and the long-term consequences that can lead to CAI [13,14]. Furthermore, economic analyses indicate that CAI incurs not only substantial direct costs, including healthcare expenditures, but also significant indirect costs, such as lost work productivity, collectively imposing a considerable burden on the society [15].

Recent studies have suggested that local ankle instability can influence the central functions of the trunk and hip, potentially affecting core muscle activation and balance control [16,17]. Of particular interest is the diaphragm, which serves not only as the primary muscle of inspiration but also plays a key role in regulating intra-abdominal pressure and maintaining trunk stability [18,19]. The diaphragm coordinates with deep core muscles such as the multifidus, transversus abdominis, and pelvic floor muscles to control posture and stabilise proximal muscle groups [20,21]. Previous research has demonstrated that diaphragmatic function affects postural control by regulating intra-abdominal pressure and enhancing trunk stability [18,22,23]. Dysfunction of this system has been linked to impaired motor control, reduced diaphragm contractility, and the potential risks of musculoskeletal instability and injury, as observed in individuals with CAI and increased respiratory demand [18,23].

However, postural dysfunction in individuals with CAI may also be influenced by other factors, such as poor postural habits, overuse injuries, or activity-specific loading patterns. Additionally, neuromuscular control may vary by age and sex, which could influence both diaphragm and hip flexor function. These considerations highlight the multifactorial nature of postural deficits in CAI.

Research has indicated that individuals with CAI may exhibit reduced diaphragmatic function, which may contribute to impaired postural control [24,25]. Therefore, assessments and interventions for CAI should address not only the muscles surrounding the ankle but also the function of the deep core muscles, including the diaphragm.

Anatomically, the diaphragm is directly connected to the iliopsoas (comprising the psoas major and iliacus), suggesting a potential bidirectional relationship between these muscles [26,27]. As the primary hip flexor, the iliopsoas is essential for walking, running, and maintaining posture [28,29,30,31,32]. If impaired diaphragmatic function inhibits iliopsoas activation, it may increase the risk of re-spraining or impair athletic performance in individuals with CAI. The hypothesised mechanism underlying this relationship is illustrated in Figure 1.

This study investigated the effects of diaphragmatic function on hip flexor activity in patients with CAI. Specifically, this study seeks to answer the following questions:(i)Iliopsoas activity between the affected (sprained) and contralateral (control) sides during maximal isometric hip flexion under three breath-holding conditions: end-expiration, end-inspiration, and intermediate state.(ii)The activity and torque output of the other hip flexor muscles (rectus femoris, sartorius, and tensor fasciae latae) were measured to determine whether a reduction in iliopsoas activity affected force production.(iii)Propose a comprehensive rehabilitation strategy for individuals with CAI based on these findings.

This study aimed to bridge the gap in the existing research on neuromuscular control deficits in CAI and provide a foundation for future clinical applications. Specifically, incorporating diaphragm function assessments into CAI evaluations may refine rehabilitation strategies by addressing their impact on hip flexor activity. By examining the interaction between diaphragmatic function and iliopsoas activation, this study aimed to enhance our understanding of musculoskeletal injuries, emphasising both local joint mechanics and central neuromuscular coordination in rehabilitation approaches. Clarifying the relationship between diaphragmatic function and hip flexor activation may contribute to the development of targeted rehabilitation strategies. Incorporating diaphragmatic training into CAI rehabilitation programmes can improve functional stability, reduce re-injury rates, and enhance athletic performance.

## 2. Results

### 2.1. Hip Flexor Muscle Activity

Electromyographic (EMG) signals from the four hip flexor muscles showed that the root mean square (RMS) value of the iliopsoas on the affected side was significantly lower than that on the control side during end inspiration (*p* < 0.05). This finding suggests that diaphragmatic activity during inspiration may influence iliopsoas activation in patients with CAI. In contrast, no significant differences were observed between the sides during end-expiration and intermediate conditions (*p* > 0.05), indicating that the impact of breath-holding on iliopsoas activation may be phase-dependent. Additionally, no significant differences were found between the affected and control sides for the rectus femoris, sartorius, and tensor fasciae latae under any breath-holding condition (*p* > 0.05), suggesting that this effect was limited to the iliopsoas rather than affecting all hip flexors.

Table 1 presents the detailed RMS values for each muscle under different breath-holding conditions on both the affected and control sides. Figure 2 compares the RMS values of the hip flexor muscles under different breath-holding conditions on both the affected and control sides. However, no significant differences in RMS values were observed across the breath-holding conditions on either side.

The table presents the root mean square (RMS) values of the four hip flexor muscles—iliopsoas (IL), rectus femoris (RF), sartorius (SA), and tensor fasciae latae (TFL)—measured under three breath-holding conditions: maximum expiration (ME), maximum inspiration (MI), and an intermediate position (IP). The values are reported as mean ± standard deviation. A significant difference in iliopsoas RMS values was observed between the affected and healthy sides during inspiration, with the healthy side showing significantly higher values (*p* < 0.05). However, no significant differences were found between the affected and healthy sides for the other hip flexor muscles under any breath-holding condition. These findings suggest that diaphragmatic activity during inspiration may selectively influence iliopsoas activation, whereas other hip flexors remain unaffected. These data provide insights into muscle activation patterns and the potential influence of the respiratory state on hip flexor function in individuals with chronic ankle instability.

The bar graph illustrates the RMS values of the four hip flexor muscles—iliopsoas (IL), rectus femoris (RF), sartorius (SA), and tensor fasciae latae (TFL)—measured under three breath-holding conditions: maximum expiration (ME), maximum inspiration (MI), and an intermediate position (IP). Separate comparisons were conducted for each muscle across breath-holding conditions on the healthy and affected sides. However, no significant differences in RMS values were found across breath-holding conditions for any of the muscles on either side. These findings suggest that the breath-holding state does not significantly alter hip flexor muscle activation in individuals with chronic ankle instability.

### 2.2. Torque Measurements

The peak isometric hip flexion torque did not differ significantly between the affected and control sides under any of the three breath-holding conditions (*p* > 0.05). Additionally, no significant within-group differences were found when comparing the torque values across breath-holding conditions on either side. Figure 3 shows the detailed torque values, demonstrating that the respiratory state has no measurable impact on the maximal hip flexion strength in individuals with CAI. These findings suggest that, unlike muscle activation patterns, peak torque generation remains consistent regardless of the breath-holding condition.

The line graph depicts torque values for both sides across various breath-holding conditions: maximum expiration (ME), maximum inspiration (MI), and an intermediate position (IP). No significant differences in peak isometric hip flexion torque were observed between the affected and healthy sides under any condition (*p* > 0.05). Additionally, within-group comparisons revealed no significant variations in torque across breath-holding conditions for either side. These findings suggest that the respiratory state does not influence maximal hip flexion strength in individuals with chronic ankle instability, in contrast to its potential effect on muscle activation patterns.

## 3. Discussion

This is the first study to evaluate the effect of breath-holding conditions on iliopsoas activity in individuals with CAI, revealing a significant reduction in muscle activity on the affected side during end-inspiration. This finding suggests that impaired diaphragmatic function may suppress iliopsoas activity, potentially contributing to the altered neuromuscular coordination in the lumbopelvic region. 

### 3.1. Relationship Between Diaphragm Function and Iliopsoas Activity

The diaphragm plays a key role not only in respiration but also in maintaining intra-abdominal pressure and trunk stability [22,33]. During end inspiration, the diaphragm is nearly maximally contracted, requiring coordinated action with the deep core muscles, including the multifidus, transversus abdominis, and pelvic floor muscles [20,34]. Optimal coordination between these muscles is essential for stabilising the lumbopelvic region and facilitating efficient movement. In individuals with CAI, prolonged pain avoidance and joint instability may disrupt this motor chain, impairing deep core muscle control [35,36,37]. These impairments can lead to compensatory movement patterns, reduced postural stability, and inefficient force transmission during functional tasks. The observed reduction in iliopsoas activity on the affected side during end-inspiration suggests that impaired diaphragmatic contraction and compromised intra-abdominal pressure regulation may underlie this phenomenon. 

Additionally, given its anatomical connection to the diaphragm [26,27], weakened iliopsoas may further contribute to movement asymmetry and reduced functional performance. This finding suggests a possible association between diaphragmatic function and reduced iliopsoas activity in individuals with CAI. However, as this study did not directly measure diaphragmatic performance, this interpretation remains speculative. Further investigation using direct assessments of diaphragmatic motion and contractility is necessary to determine a causal relationship. 

### 3.2. Comparison with Other Hip Flexor Muscles and Impact on Force Production

No significant differences were found between the affected and control sides for the rectus femoris, sartorius, and tensor fasciae latae across all breath-holding conditions, and no differences were observed in the hip flexion torque. This finding suggests that secondary hip flexors may compensate for reduced iliopsoas activation, thereby maintaining total torque output during static contractions [38,39]. However, since muscle activation does not always translate into measurable strength changes, the clinical significance of reduced iliopsoas EMG activity should be interpreted with caution. Further research is needed to examine whether such neuromuscular adaptations influence dynamic tasks or contribute to long-term musculoskeletal issues. While no differences in peak torque were observed under static conditions, diminished iliopsoas activity may have more substantial effects during dynamic, multi-joint movements such as stair climbing or sports-specific tasks [32,40,41,42]. Over time, the compensatory recruitment of the rectus femoris, sartorius, and tensor fasciae latae may lead to muscular imbalances and disrupted coordination in the lower limb, potentially increasing the risk of further musculoskeletal issues. Further research is needed to examine how such neuromuscular adaptations affect performance and stability in functional and sport-related activities.

### 3.3. Comparison with Clinical Implications and Rehabilitation Applications

Our findings provide new insights into the rehabilitation of CAI. Traditional rehabilitation approaches typically focus on strengthening the muscles surrounding the ankle and improving balance [43,44]. However, impaired coordination of the diaphragm and deep core muscles [24,25] may also contribute to reduced activity or the coordination of proximal muscles such as the iliopsoas [45,46,47]. This dysfunction can lead to decreased lumbopelvic stability, inefficient force transfer during movement, and compensatory patterns, which may further compromise lower-limb mechanics.

Clinically, incorporating diaphragmatic breathing exercises to enhance diaphragmatic function and intra-abdominal pressure regulation, combined with core stability training targeting the multifidus, transversus abdominis, and pelvic floor muscles, may improve functional connectivity between the diaphragm and iliopsoas. Restoring proper neuromuscular control in the lumbopelvic region may help optimise postural stability, reduce compensatory muscle activation, and improve movement efficiency in individuals with CAI. 

Additionally, functional training aimed at improving coordination between the diaphragm and iliopsoas during dynamic activities such as walking, running, and cutting manoeuvres may help reduce the risk of re-sprains and enhance movement efficiency. To enhance clinical reproducibility, it is essential that these training strategies be organised into structured, progressive, and quantifiable protocols. The use of standardised exercise parameters—such as duration, intensity, and breathing tempo—alongside objective outcome measures, including EMG activation, torque generation, and balance performance, will facilitate more reliable comparisons and replication across studies and clinical settings. Dynamic exercises that integrate breathing techniques with lower-limb movements, such as resisted breathing drills during gait training or plyometric activities, may be effective in reinforcing this neuromuscular connection. Furthermore, incorporating breathing control strategies into sports-specific drills, such as acceleration–deceleration tasks or change-of-direction exercises, may further optimise functional performance and prevent injury.

Appropriate movement-based assessments of diaphragmatic function are essential to assess the effectiveness of these interventions and better understand their underlying mechanisms. Movement-based assessments of diaphragmatic function, such as inspiratory muscle testing, diaphragmatic ultrasonography, and spirometry, could provide deeper clinical insights into their contribution to lower-limb stability and neuromuscular control. Combining these assessments with EMG analysis of the iliopsoas and other hip stabilisers would allow for a more comprehensive evaluation of the interplay between respiratory mechanics and proximal muscle activation.

### 3.4. Limitations

This study has certain limitations. Although the sample size in this study was relatively small, which limits the generalizability of the findings, it serves as a preliminary exploration that offers insights for future research. As a pilot study, it provides foundational data that can inform the design of larger, more comprehensive investigations. In addition, the age range of participants was relatively narrow, which further limits the external validity of the findings. Given that neuromuscular characteristics may vary substantially across age groups, especially among younger and older individuals, future studies should include participants with a wider range of ages and physical activity levels to improve generalizability. To strengthen the validity and reliability of future findings, subsequent studies should include larger samples to enhance statistical power and allow for subgroup analyses. In support of this, we conducted a post hoc power analysis based on the observed effect size for iliopsoas activity during the end-inspiration condition (Cohen’s d = 1.04). With α = 0.05 and a paired *t*-test model, the estimated minimum sample size to achieve a power of 0.80 was eight participants. Since our study included 11 participants, the statistical power was sufficient to detect this effect. This supports the robustness of our main findings and provides reference values for future study design. As the study only assessed static maximal isometric contractions, the findings may not fully reflect the demands of real-life dynamic activities.

Furthermore, we acknowledge that respiratory states were not confirmed using imaging modalities such as diaphragmatic ultrasound or MRI. Including these objective techniques in future studies would improve the accuracy of respiratory phase classification and strengthen the mechanistic interpretation of the observed neuromuscular responses. In this study, diaphragmatic function was inferred indirectly from breath-holding conditions, without direct measurement. The absence of imaging-based assessments limits our ability to substantiate the proposed mechanism. To address this limitation, future research should include direct evaluations of diaphragmatic motion and contractility to validate the underlying hypothesis.

This study focused primarily on the iliopsoas as a representative proximal muscle, given its anatomical and functional relationship with the diaphragm. However, chronic ankle instability (CAI) is known to disrupt neuromuscular control throughout the kinetic chain. It affects not only proximal stabilisers, but also distal and synergistic muscle groups. While the current findings provide insight into the local interaction between diaphragmatic function and hip flexor activity, they offer a limited perspective. They do not fully capture the broader neuromuscular adaptations associated with CAI, highlighting the need for future research to explore these mechanisms more comprehensively.

### 3.5. Future Directions

Future research should investigate the interaction between the diaphragm and iliopsoas during dynamic tasks and assess the role of diaphragmatic function across a broader range of activities. For instance, studies exploring diaphragmatic engagement during high-intensity locomotor tasks, such as sprinting, sudden acceleration, deceleration, or agility drills, could provide deeper insight into its role in motor control and movement efficiency. Investigating the activation patterns of additional core and lower-extremity muscles, as well as evaluating balance and postural control during functional activities, would provide a more comprehensive understanding of motor adaptations in CAI. Examining its contribution to stability during unilateral stance or load-bearing activities would further clarify its impact on lower-limb function. Additionally, integrating real-time respiratory monitoring with EMG and motion capture analysis would offer a more comprehensive understanding of how breathing patterns influence hip flexor activation and whether altered diaphragmatic function leads to compensatory movement strategies.

Furthermore, randomised controlled trials are necessary to evaluate the effectiveness of rehabilitation programmes aimed at enhancing diaphragmatic function in patients with CAI. These trials should adhere to reproducible and standardised intervention protocols that include clearly defined criteria for exercise progression, reliable tools for monitoring respiratory engagement, and the use of validated outcome measures to assess effectiveness. Interventions incorporating diaphragmatic breathing exercises, intra-abdominal pressure training, and functional movement drills should be examined to determine their effects on neuromuscular coordination, injury prevention, and athletic performance. Future trials should also assess individualised training protocols to determine whether tailored diaphragmatic interventions yield superior outcomes compared with generalised rehabilitation approaches. Including control groups following standard rehabilitation protocols would facilitate direct comparison and provide clear evidence of their efficacy.

Longitudinal studies assessing whether improvements in diaphragmatic function correlate with reduced re-sprain rates and enhanced functional outcomes would further strengthen the clinical relevance of this study. Additionally, investigating the potential neuromechanical link between diaphragmatic function and spinal alignment, pelvic stability, and limb symmetry may provide further insight into broader musculoskeletal implications. Exploring whether enhanced diaphragmatic function contributes to improved proprioception, balance, and postural stability could broaden its applicability beyond CAI rehabilitation to musculoskeletal health. Finally, interdisciplinary research combining sports science, biomechanics, and respiratory physiology could help to develop a more integrative framework for optimising movement performance and injury prevention.

## 4. Materials and Methods

### 4.1. Relationship Between Diaphragm Study Design and Participants

This cross-sectional observational study investigated the relationship between breath-holding conditions and hip flexor muscle activity in patients with a history of chronic ankle sprain (CAS). This study specifically targeted adults diagnosed with functional ankle instability, which was confirmed through clinical evaluation using the Cumberland Ankle Instability Tool and the anterior drawer test.

A total of eleven participants (six males, five females; mean age: 28.5 ± 4.2 years) were recruited based on the following inclusion criteria:A history of CAS with onset at least six months prior to the study.Subjective instability reported in one ankle.A clear distinction between affected and unaffected limbs, allowing for side-to-side comparisonsNo history of ankle surgery or other major lower-limb interventions in the past year.Engagement in moderate or high levels of physical activity was defined as participation in sports or recreational exercise at least once a week.No history of other lower-limb musculoskeletal conditions, such as knee or hip disorders.

All procedures were conducted in accordance with the ethical standards outlined in the Declaration of Helsinki and approved by the Kanazawa Orthopaedic Sports Medicine Clinic Ethics Committee (approval number: Kanazawa-OSMC-2020-004). Written informed consent was obtained from all participants before the commencement of the study.

### 4.2. Muscle Measurement and Equipment

#### 4.2.1. sEMG

sEMG signals were recorded from four hip flexor muscles: the iliopsoas (IL), rectus femoris (RF), sartorius (SA), and tensor fasciae latae (TFL). Active electrodes (MQ8/16 16-bit EMG amplifier; Kissei Comtec, Nagano, Japan), measuring 1.0 × 1.0 mm, were applied with an interelectrode distance of 10 mm. The high impedance of the electrodes ensured accurate bioelectrical signal capture even during dynamic movements. The system digitised the signals near the electrodes, minimised noise, and reduced preprocessing time. The EMG signals were recorded at a sampling rate of 2000 Hz using a telemetry system (MQ16; Kissei Comtec) and analysed using a dedicated software (Kine Analyser; Kissei Comtec). During each trial, muscle activity was recorded continuously for 5 s. 

#### 4.2.2. Electrode Placement

Electrodes were placed over specific anatomical landmarks to ensure the accurate measurement of muscle activity. Each muscle was placed as follows:Iliopsoas: 3–5 cm distal to the anterior superior iliac spine (ASIS), with ultrasound guidance used to confirm proper subfascial placement [48,49].Rectus femoris: midpoint between the anterior inferior iliac spine (AIIS) and upper edge of the patella.Sartorius: Along the line connecting the ASIS and medial tibial condyle, approximately 8 cm distal to the ASIS.Tensor fasciae latae: Midpoint between the ASIS and apex of the greater trochanter.

The electrodes were aligned with the muscle fibres, and a reference electrode was placed on the right patella to ensure consistency. Before electrode placement, the skin was shaved, cleaned with alcohol, and lightly abraded to reduce impedance and enhance the signal quality.

#### 4.2.3. Isokinetic Dynamometer

A Cybex isokinetic dynamometer was used to measure the peak torque during maximal isometric hip flexion. The participants were placed in the supine position with the hip at 0° and the knee flexed at 90° to effectively isolate the hip flexor muscles. Adjustable stabilisation straps were applied over the trunk and pelvis to prevent compensatory movements and to ensure accurate force measurements.

#### 4.2.4. Experimental Conditions (Breath-Holding Conditions)

Muscle activity and torque were measured under three distinct breath-holding conditions:End expiration (functional residual capacity state).End inspiration (maximum lung inflation state).Intermediate state (resting expiration level).

For each condition, the participants performed a 5 s maximal isometric hip flexion, with measurements recorded for both the affected (sprain history) and control limbs.

The sequence of breath-holding conditions was randomised for each participant to ensure randomisation and to prevent order effects. A rest period of 3 min was provided between trials to prevent fatigue and maintain consistency in muscle performance.

A spirometer was used in each trial to verify the specific breath-holding phase—end-expiration, end-inspiration, or intermediate—while oxygen saturation and heart rate were continuously monitored to ensure physiological consistency across participants throughout the assessments. A spirometer was used to monitor respiratory patterns to verify adherence to breathing conditions, and oxygen saturation and heart rate were continuously monitored for safety purposes.

### 4.3. Data Analysis

#### 4.3.1. EMG Analysis

The EMG signals were full-wave rectified and processed using a bandpass filter with a frequency range of 10–500 Hz to reduce noise and improve signal fidelity.

For each trial, the central 3 s period during which the participant maintained a stable maximal effort was extracted for analysis. The root mean square (RMS) value of the EMG signal, which serves as an indicator of muscle activation, was calculated.

The RMS values measured during the intermediate breathing condition were normalised by designating them as 100%, and the RMS values obtained under the end-expiratory and end-inspiratory conditions were expressed as percentages relative to this reference. This normalisation approach ensured that individual variations in absolute muscle activity did not confound comparative analysis.

#### 4.3.2. Torque Analysis

The peak torque values recorded during each 5 s maximal isometric contraction were extracted and compared across the three breath-holding conditions. Additionally, torque differences between the affected and control limbs were examined to assess the impact of breathing patterns on force generation.

#### 4.3.3. Statistical Analysis

A paired *t*-test was used to compare the muscle activity and torque between the affected and control sides for each breath-holding condition. Differences across breath-holding conditions were assessed using repeated-measures analysis of variance (ANOVA) with Bonferroni post hoc tests. Statistical significance was set at *p* < 0.05. All analyses were performed using the SPSS software (IBM SPSS Statistics 27).

## 5. Conclusions

This study investigated the influence of different breath-holding conditions on iliopsoas and other hip flexor muscle activities in individuals with CAI. Given the close anatomical and functional relationship between the diaphragm and deep core muscles, including the iliopsoas, understanding how respiratory mechanics influence lower-limb muscle activation is crucial for optimising rehabilitation strategies.

The results showed that during the end-inspiration condition, iliopsoas muscle activity on the affected side was significantly lower than that on the control side (*p* < 0.05). In contrast, no significant differences were observed in the activation levels of the other hip flexor muscles (rectus femoris, sartorius, and tensor fasciae latae) between the affected and control sides under any breathing conditions. Additionally, no significant differences were found in peak hip flexion torque across breath-holding conditions.

These findings suggest a potential association between diaphragmatic function and iliopsoas activity in individuals with CAI. However, given that this study utilised surface EMG under static, isometric conditions, the interpretation of neuromuscular coordination should be approached with caution. The influence of diaphragmatic function under dynamic, functional movements remains to be clarified.

From a clinical perspective, the findings tentatively support the consideration of incorporating core-focused interventions that include diaphragmatic engagement into rehabilitation programmes. Nevertheless, further research using dynamic assessment methods and imaging tools is needed to validate these preliminary observations.

Future studies should employ tools such as diaphragmatic ultrasound, spirometry, or dynamic MRI to evaluate diaphragmatic function more directly, and intervention trials are needed to determine the efficacy of breathing-based training programmes in improving lower-limb neuromuscular control.

In summary, while this pilot study offers early insights into the possible link between breathing patterns and proximal muscle activity in CAI, the findings should be interpreted within the methodological limits of static surface EMG assessment.

## Figures and Tables

**Figure 1 muscles-04-00016-f001:**
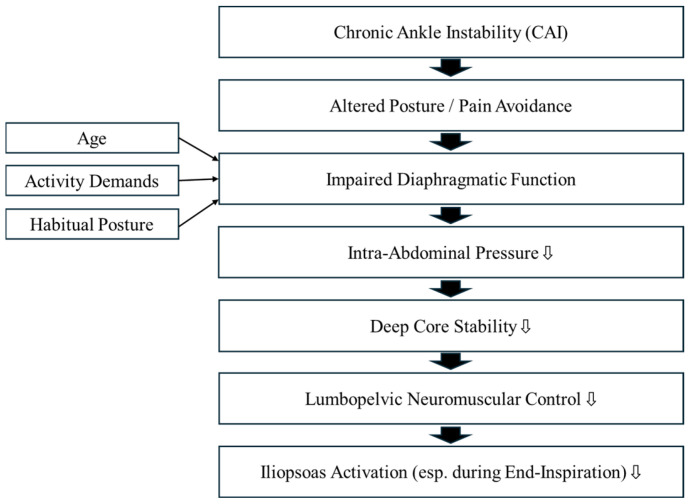
Hypothesised mechanism linking diaphragmatic function to iliopsoas activation in individuals with chronic ankle instability (CAI).

**Figure 2 muscles-04-00016-f002:**
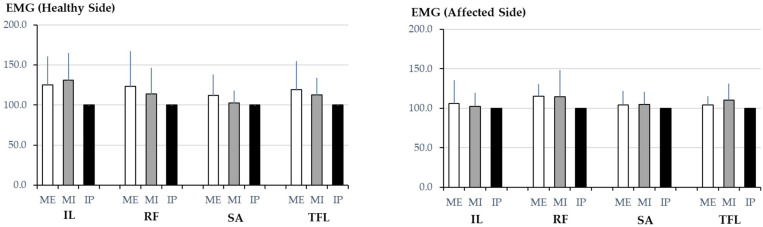
Comparison of root mean square (RMS) values of hip flexor muscles across different breath-holding conditions for the healthy and affected sides.

**Figure 3 muscles-04-00016-f003:**
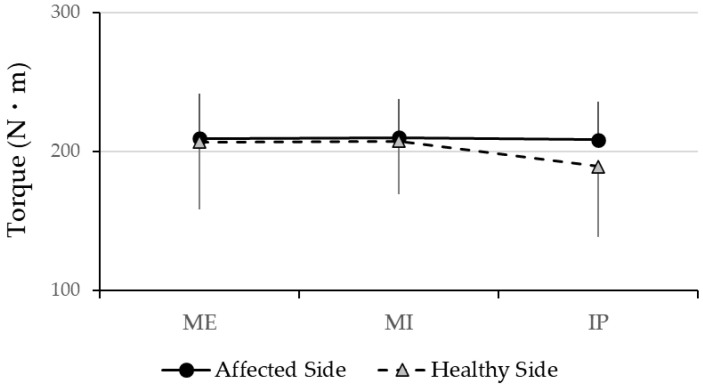
Comparison of maximum isometric hip flexion torque under different breath-holding conditions between the healthy and affected sides.

**Table 1 muscles-04-00016-t001:** RMS values of hip flexor muscles on the healthy and affected sides under different breath-holding conditions (mean ± standard deviation).

Muscle	Condition	Healthy Side (*n* = 11)	Affected Side (*n* = 11)	*p*-Value
Iliopsoas	Maximum inspiration	131 ± 34	103 ± 17	0.014 *
	Maximum expiration	125 ± 36	106 ± 29	0.129
	Intermediate position	100 ± 0	100 ± 0	—
Rectus Femoris	Maximum inspiration	114 ± 33	115 ± 16	0.939
	Maximum expiration	132 ± 67	115 ± 33	0.372
	Intermediate position	100 ± 0	100 ± 0	—
Sartorius	Maximum inspiration	103 ± 15	105 ± 16	0.792
	Maximum expiration	112 ± 26	104 ± 18	0.361
	Intermediate position	100 ± 0	100 ± 0	—
Tensor Fasciae Latae	Maximum inspiration	113 ± 21	110 ± 21	0.673
	Maximum expiration	119 ± 36	104 ± 11	0.212
	Intermediate position	100 ± 0	100 ± 0	—

* Statistically significant difference (*p* < 0.05).

## Data Availability

The data presented in this study are available on request from the author.

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
