# Peer review of "A Pilot Study on the Influence of Diaphragmatic Function on Iliopsoas Muscle Activity in Individuals with Chronic Ankle Instability"

_muscles, 2025, doi:10.3390/muscles4020016_

Round 1

Reviewer 1 Report

Comments and Suggestions for Authors

General Assessment:

This study presents an interesting and novel investigation into the relationship between respiratory mechanics—specifically breath-holding states—and hip flexor muscle activity in individuals with chronic ankle instability (CAI). The hypothesis that diaphragmatic dysfunction may influence iliopsoas activation and thereby contribute to proximal motor control deficits is both timely and relevant, given the growing recognition of the integrated nature of the core and lower extremity neuromuscular systems. While the study offers valuable preliminary findings, several methodological limitations and interpretational considerations merit attention.

Major Comments:

  1. Sample Size: The sample size (n=11) is small and may limit the generalizability of the results, and the authors point that out as a caveat as well. The ability to detect subtle but clinically meaningful differences is constrained, and the risk of Type I and II errors remains high. Future studies should aim to recruit a larger cohort to improve statistical robustness and enable subgroup analyses.

  2. Validity of the Testing Paradigm: The exclusive use of maximal isometric hip flexion tasks limits the functional relevance of the findings. CAI typically affects individuals during dynamic and weight-bearing activities. Thus, assessing muscle activation during gait, balance tasks, or sport-specific movements would provide more ecologically valid insights into neuromuscular control in this population.

  3. Respiratory Conditions: The breath-holding conditions—end-expiration, end-inspiration, and intermediate—are conceptually sound but lack objective verification. Without physiological measures (e.g., spirometry, ultrasound imaging of the diaphragm), it is difficult to confirm that participants achieved and maintained the intended respiratory state across trials. 

  4. Diaphragmatic Function: While the authors posit that reduced iliopsoas activity during end-inspiration reflects impaired diaphragmatic function, this link remains speculative in the absence of direct assessments of diaphragm performance. Including imaging or functional tests of diaphragmatic movement would strengthen the argument for a mechanistic relationship. Also, diaphragmatic movement will affect blood oxygen that could in turn affect the iliopsoas activity as well.

  5. Interpretation of Torque Data: Peak hip flexion torque did not differ significantly between the affected and control limbs, or across breath-holding conditions. This raises questions about the functional implications of the reduced iliopsoas EMG activity. The clinical relevance of isolated changes in muscle activity, absent corresponding deficits in force output, should be interpreted with caution.

  6. Muscle Analysis: While the study focuses on the iliopsoas, CAI is associated with altered neuromuscular control across the kinetic chain. Broader assessment of synergistic and stabilizing muscles, as well as postural and balance control strategies, would offer a more comprehensive understanding of motor adaptation in this population.

Comments on the Quality of English Language

English can be improved for clarity

Reviewer 2 Report

Comments and Suggestions for Authors

The authors perform a unique study to examine the influence of diaphragm function on ankle sprain / ankle instability.   While the sample sizes were small, which likely led to under-sampled statistical insignificance,  it is worthy study and thought experiment.

The authors should include a figure the explains the study and shows a mechanism/hypothesis were WHY they think there could be differences in diaphragm function between individuals.

Add x and y axis lines to Figure 1 bar graphs and Figure 2.

Perform a power analysis using the effect size and variance of this study to determine how many patients are needed to assess possible significance,.

Reviewer 3 Report

Comments and Suggestions for Authors

This study was of interest because of its high incidence.

Introduction

Despite the indisputable role of the diaphragm, other types of postural deficits due to other casuistry (postural atutudity, overuse injuries, type of physical activity, etc.) should be mentioned, the authors make the diffragmatic as the only cause, they should try to have a less partial vision.

They have not discussed the main reasons that can trigger this, nor the incidence by sex/age; it could be a little more nuanced.

Methods

Due to the number of people, it is strongly suggested that the title be changed to a pilot study.

The form of EMG analysis should be described, and only the device is described.

Discussion

The argument for future applications should be strengthened so that the reader can see reproducibility.

In the limitations, it should be pointed out that the sample is not only small but also of a very specific age, which makes generalization impossible.

Conclusions

Please tone down the findings, since as you yourselves indicate, EMG is superficial and isometric, without the possibility of being able to argue all contexts.
